# STORM: Benchmarking Visual Rating of MLLMs with a Comprehensive Ordinal Regression Dataset

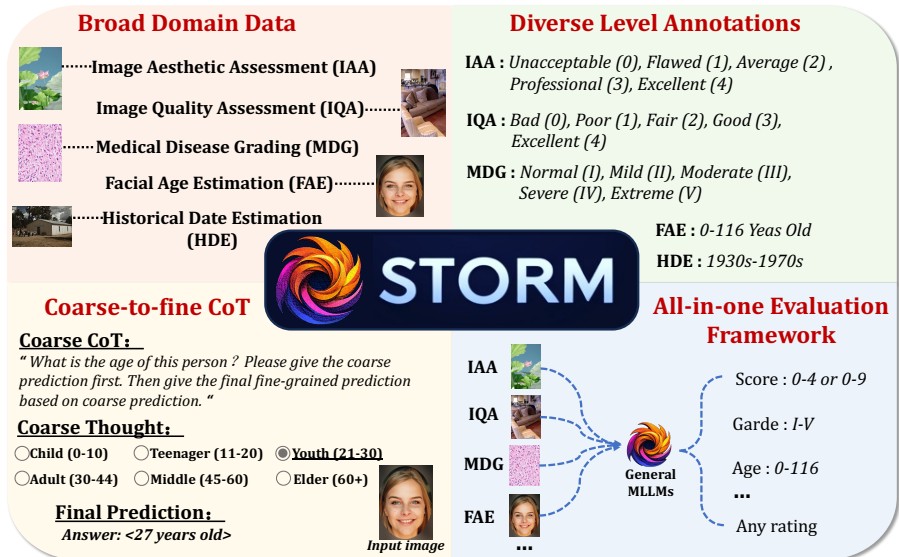

Figure 1: An overview of our STORM benchmark. STORM consists of four key components: 1) Broad domain data (14 datasets across 5 domains); 2) diverse level annotations; 3) coarse-to-fine CoT; 4) all-in-one visual rating framework.

## Abstract

Visual rating is an essential capability of artificial intelligence (AI) for multi-dimensional quantification of visual content, primarily applied in ordinal regression (OR) tasks such as image quality assessment, facial age estimation, and medical image grading. However, current multi-modal large language models (MLLMs) under-perform in such visual rating ability while also suffering the lack of relevant datasets and benchmarks. In this work, we collect and present STORM, a data collection and benchmark for **S**timulating **T**rustworthy **O**rdinal **R**egression Ability of **M**LLMs for universal visual rating. STORM encompasses 14 ordinal regression datasets across five common visual rating domains, comprising 655K image-level pairs and the corresponding carefully curated VQAs. Importantly, we also propose a coarse-to-fine processing pipeline that dynamically considers label candidates and provides interpretable thoughts, providing MLLMs with a general and trustworthy ordinal thinking paradigm. This benchmark aims to evaluate the all-in-one and zero-shot performance of MLLMs in scenarios requiring understanding of the essential common ordinal relationships of rating labels. Extensive experiments demonstrate the effectiveness of our framework and shed light on better fine-tuning strategies. The STORM datasets and codes are available on anonymous links in the Supplementary Material to support further research in this area.

## 1 Introduction

With the success of large language models (LLMs) like GPT-4 Achiam et al. (2023a) and Gemini Team et al. (2023), researchers have been enhancing these models by incorporating visual understanding ca-

pabilities. This enthusiasm has led to the emergence of multi-modal large language models (MLLMs), such as LLaVA Liu et al. (2023a;b), GPT-4o Achiam et al. (2023b), and Qwen-VL Bai et al. (2023a;b), which show demonstrated viability in various VQA scenarios.

However, the potential of MLLMs in visual rating capabilities has not yet been fully explored despite their critical importance in various visual analysis applications, such as image quality/aesthetic assessment, face age estimation, medical image grading, etc. The hindrance in the development of stronger MLLMs for visual rating is attributed to the following three challenges. (1) The complexity of task labels, that is, inconsistent numbers and levels of labels of different visual rating tasks. Existing methods only train MLLMs with the same number and definition of level labels Wu et al. (2023c), which could yield unsatisfied performance when users propose a different rating protocol. (2) The hallucination phenomenon of MLLMs for numeric labels. MLLMs typically use contrastive learning for pre-training and pay more attention to high-level semantics than to precise numerical features Wu et al. (2023b). Furthermore, the subjective inconsistency of human annotation can also lead the model to learn noise. (3) Poor zero-shot performance. Existing MLLMs can only be trained on specific tasks, which can incur severe limitations when the model is tested on out-of-domain datasets and may lack general rating practicality. Unfortunately, there is still a lack of relevant datasets and benchmarks to train and test trustworthy MLLMs with strong and general visual rating capabilities.

To address the above challenges, we look into the inherent logic of common visual rating tasks and observe a shared nature of these tasks: They are all ordinal regression (OR) problems whose labels are ordinal. Therefore, we introduce STORM, a data collection and benchmark for **S**timulating **T**rustworthy **O**rdinal **R**egression Ability of **M**LLMs for universal visual rating. First, STORM includes a comprehensive OR data collection comprising 655K question-answer pairs across 5 popular visual rating tasks. Through joint training based on this comprehensive OR dataset, an MLLM is initially endowed with a fundamental ability to tackle most visual rating tasks. Furthermore, we develop a lite version dataset of about 250K samples for faster model training. Second, for all question-answer pairs, the answer not only adopts a mixed description of text and numbers to significantly mitigate the model's numeric hallucination but also includes an extra intermediate prediction step, which is designed to instruct the MLLM with a logical, coarse-to-fine Chain-of-Thought (CoT) process to understand a general way of thinking about OR problems, enabling MLLMs to attain a better zero-shot performance on out-of-domain visual rating tasks. Third, we provide the corresponding visual rating benchmark and pre-trained models for reproducibility, aiming to foster further research in visual rating for MLLMs.

In summary, the key highlights of our STORM benchmark include:

- Broad Domain Data: STORM contains high-quality data including 14 popular ordinal regression datasets comprising 655k data items across five distinct domains.
- Diverse Level Annotations. STORM includes basic numeric labels, suitable for fundamental settings of all visual rating questions. It also incorporates diverse text labels to strengthen the specific semantic understanding for different visual rating tasks and the capabilities of MLLMs in explainable rating predictions.
- Coarse-to-fine CoT: We introduce a coarse-to-fine Chain-of-Thought (CoT) pipeline for MLLMs, enabling them to learn a universal paradigm of ordinal regression and providing intermediate interpretable thoughts.
- All-in-one Evaluation Framework: We propose a comprehensive evaluation framework to benchmark the all-in-one visual rating capability of MLLMs on both in-domain and out-of-domain datasets. To the best of our knowledge, STORM is the first benchmarking and dataset building effort to test the universal visual rating abilities of MLLMs.

## 2 RELATED WORKS

**Multi-modal LLMs.** The success of large language models (LLMs) in various language applications has paved the way for the development of multi-modal large language models (MLLMs), which integrate vision and language modalities. More recently, MLLMs have focused on aligning these modalities through extensive training on image-caption pairs or image-question conversations. Notable methods like LLaVA Liu et al. (2023b) train a projector that maps image tokens to aligned representations of pre-trained LLMs. Other approaches, such as BLIP-2 Li et al. (2022a; 2023a),

Table 1: A summary of the ordinal regression datasets in STORM for visual rating. STORM spans 5 domains and includes various source datasets, offering a broad representation of visual data styles.

| Domain | Source Dataset | Full Version Size | Lite Version Size | Category |
|---|---|---|---|---|
| Image Quality Assessment (IQA) | SPAQ Fang et al. (2020) | 11,125 | 11,125 | 5 levels |
| | ChallengeDB Ghadiyaram & Bovik (2015) | 1,169 | 1,169 | 5 levels |
| | KonIQ Hosu et al. (2020) | 10,073 | 10,073 | 5 levels |
| Image Aesthetics Assessment (IAA) | Aesthetics Dataset Dosovitskiy et al. (2020) | 13,706 | 13,706 | 5 levels |
| | TAD66K He et al. (2022) | 66,327 | 27,132 | 5 levels |
| | AVA Gu et al. (2018) | 255,508 | 51,104 | 5 levels |
| Facial Age Estimation (FAE) | Adience Levi & Hassner (2015) | 17,321 | 17,321 | 8 groups |
| | CACD Chen et al. (2015) | 163,446 | 32,690 | 14-62 years |
| | Morph Jr. & Tesafaye (2006) | 50,015 | 20,006 | 16-77 years |
| | UTK Zhang et al. (2017) | 24,106 | 24,106 | 1-116 years |
| Medical Disease Grading (MDG) | Eyepacs Dugas et al. | 35,127 | 35,127 | 5 grades |
| | DeepDR Liu et al. (2022) | 2,000 | 2,000 | 5 grades |
| | APTOS Karthik & Dane (2019) | 3,662 | 3,662 | 5 grades |
| Historical Date Estimation (HDE) | HCI Palermo et al. (2012) | 1,325 | 1,325 | 5 decades |

adopt a query Transformer (Q-Former) to learn image embeddings using learnable queries after obtaining image features. In terms of training strategy, recent works Liu et al. (2023b); Bai et al. (2023a); Wang et al. (2023c); Zhu et al. (2023); Chen et al. (2022); Luo et al. (2024) commonly employ a 2-stage framework that involves pre-training and vision-text alignment. MLLMs have also been extended to various applications, including fine-grained localization Wang et al. (2024); Lai et al. (2023) such as object detection Zhang et al. (2023b), video understanding Zhang et al. (2023a); Li et al. (2023b); Chen et al. (2023), and image generation Koh et al. (2024); Qian et al. (2023).

**LMMs for Visual Rating.** Some recent studies have discussed the possibilities of adopting Large Multi-modality Models (LMMs) for visual rating/scoring. For example, Q-Bench Wu et al. (2023a) and Q-Instruct Wu et al. (2023b) proposed enabling LMMs to predict quantifiable quality scores by extracting softmax pooling results on logits of two frequent tokens (*good/poor*). Another work, Q-Align Wu et al. (2023c), systematically emulated human rating and post-processing in visual rating. However, these methods are still limited in that they focus only on (1) certain types of tasks, such as image/video quality assessment and image aesthetic assessment, and (2) fixed number of categories with poor generalization. In comparison, our STORM framework introduces a comprehensive ordinal regression dataset and VQA template paradigm that contains many other tasks across different domains for visual rating in addition to image/video quality assessment and image aesthetic assessment, such as facial age estimation, medical image grading, and image historical estimation.

**Ordinal Regression.** Given an input image, ordinal regression (OR) in computer vision aims to map the image to a rank or a continuous value. Many popular methods Rothe et al. (2018); Geng et al. (2013); Frank & Hall (2001); Li & Lin (2006); Chen et al. (2017) adopted a classification framework. Some recent studies Lim et al. (2019); Liu et al. (2019); Lee & Kim (2020); Li et al. (2021) proposed ordinal distribution constraints to exploit the ordinal nature of regression. Adding prior order knowledge to loss calculation, several methods Fu et al. (2018); Diaz & Marathe (2019) created soft labels artificially by changing the distances between categories. A few advanced methods Liu et al. (2017; 2018); Li et al. (2021); Shin et al. (2022) sorted tuples that are formed by two or three instances with ordinal categories to learn the rank information. Ord2Seq Wang et al. (2023a) proposed to transform OR tasks to sequence prediction and solve ordinal regression using autoregressive models. Recent works like OrdinalCLIP Li et al. (2022b), L2RCLIP Wang et al. (2023b), and NumCLIP Du et al. (2024) used CLIP Radford et al. (2021a) for OR tasks, focusing on improving image-text alignment. Although these deep learning (DL) methods are general and effective, they need to train separate models for different OR tasks. In comparison, our proposed STORM is a general framework built on MLLMs and aims to construct an all-in-one visual rating model.

## 3 ORDINAL REGRESSION DATA COLLECTION FOR VISUAL RATING

### 3.1 OVERVIEW

Currently, a general visual rating framework is still lacking. Existing domain-specific models are predominantly optimized for fixed-format labeling schemes, thus exhibiting poor generalization capability when encountering diverse label configurations or cross-domain scenarios. To address this

Figure 2: A data example with the original VQA compared with our coarse-to-fine CoT VQA.

gap, we curate a comprehensive OR data collection that spans five distinct domains and includes 14 various source datasets, as shown in Tab. 1. For more details on distribution, see Appendix C.

To ensure a robust foundation for different visual rating tasks, our STORM data collection deliberately integrates a diverse selection of data including image quality assessment (IQA), image aesthetic assessment (IAA), facial age estimation (FAE), medical disease grading (MDG), and image historical date estimation (HDE). These data domains are intentionally chosen to cultivate a comprehensive skill set across varied visual rating tasks. 1) IQA and IAA are the most widely demanded scenarios, which enhance MLLMs' capability in subjective qualitative judgment of quality or superiority gradation. 2) Facial age estimation aids in cognitive capabilities of objective estimation tasks with continuous and wide-ranging labels, particularly in scenarios requiring precise numerical regression like depth estimation. 3) Medical disease grading fosters the ability of severity assessment in complex scenarios, which are essential for medical and anomaly detection applications. 4) Historical date estimation develops temporal awareness of MLLMs, which is vital for time-related estimation tasks.

## 3.2 DATA GENERATION DETAILS

To gather and build a comprehensive and diverse visual rating data collection, we select 14 source ordinal regression datasets across five distinct domains. As these datasets provide only images and digital labels, they are designed with a standardized VQA paradigm by reusing their images and modifying the annotations into a textual form to enable MLLMs to undergo joint training for heterogeneous tasks of diverse domains. Specifically, each data sample originally consists of a simple question and a corresponding numeric answer. However, this paradigm can lead to numerical hallucination. Hence, we add extra domain-driven prompts and coarse-to-fine CoT to mitigate this issue. An example with the original VQA and our proposed coarse-to-fine CoT process is shown in Fig. 2. Meanwhile, we adopt the form of text + numbers for the labels to enhance semantic understanding. In the following sections, we elaborate on the VQA details employed for each domain-specific visual rating dataset.

**Image Quality Assessment (IQA).** We choose three IQA datasets to create data in this domain: SPAQ Fang et al. (2020), KonIQ Hosu et al. (2020), and ChallengeDB Ghadiyaram & Bovik (2015). The three datasets focus on the impact of distortions and other quality issues in images on human perception. The fact that these datasets provide only mean opinion score (MOS) values makes it difficult to teach LMMs to predict scores aligned with human. Thus, we simulate the process of training human annotators. We convert the MOS values to five text-defined rating levels itu (2000): {'bad' (0), 'poor' (1), 'fair' (2), 'good' (3), 'excellent' (4)}. For coarse intermediate thoughts, the candidates are: {'below fair' (0-1), 'fair' (2), 'above fair' (3-4)}.

**Image Aesthetics Assessment (IAA).** For this domain, we use Aesthetics Dataset Dosovitskiy et al. (2020), TAD66K He et al. (2022), and AVA Gu et al. (2018), which are widely-used datasets for

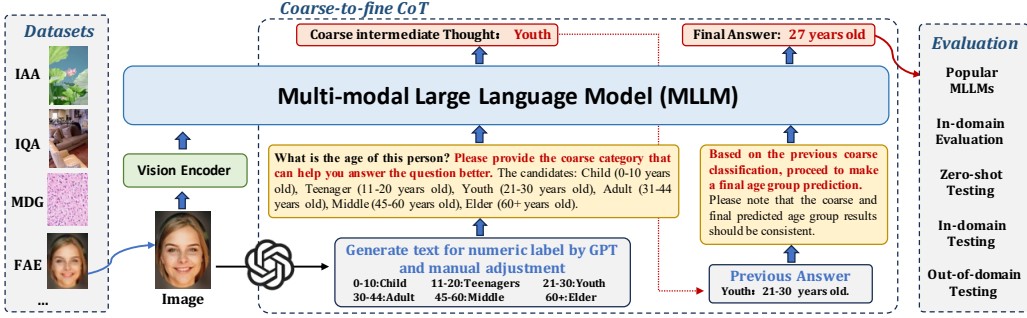

Figure 3: The model pipeline of STORM. It first extracts visual tokens of an input image from STORM dataset and determines the task objective. Then, pre-generated coarse and fine candidate categories, including numeric and text labels (generated by GPT and manually adjusted, stored in the dataset), are used to formulate instructional prompts that guide the model to perform coarse-to-fine CoT, thus predicting the corresponding labels for the image progressively. Finally, STORM conducts a comprehensive evaluation and testing, including in-domain and out-of-domain tasks.

image aesthetics assessment. The IAA datasets provide images and the corresponding multi-rater scores. Similarly to IQA, we compute the MOS values of all raters and convert the MOS values to five text-defined rating levels: {'unacceptable' (0), 'flawed' (1), 'average' (2), 'professional' (3), 'excellent' (4)}. For coarse intermediate thoughts, the candidates are: {'below average' (0-1), 'average' (2), 'above average' (3-4)}.

**Facial Age Estimation (FAE).** We use Adience Levi & Hassner (2015), CACD Chen et al. (2015), Morph Jr. & Tesafaye (2006), and UTK Zhang et al. (2017) as datasets for facial age estimation tasks. In the Adience dataset, each image encompasses a category label that is annotated in 8 groups: 0-2, 4-6, 8-13, 15-20, 25-32, 38-43, 48-53, and over 60 years old. Thus, we assign the most suitable text for each group according to the age range: {'infants' (group0, 0-2 years old), 'preschoolers' (group1, 4-6 years old), 'preteens' (group2, 8-13 years old), 'teens' (group3, 15-20 years old), 'adult' (group4, 25-32 years old), 'midlifers' (group5, 38-43 years old), 'matures' (group6, 48-53 years old), 'seniors' (group7, over 60 years old)}. In the other three datasets, each label is a specific age number. Hence, we set the answer as an age with a corresponding text, such as 'adult' (30 years old). The coarse intermediate thoughts for all these datasets are the same and the candidates are: {'baby' (group0-1, 0-7 years old), 'teenagers' (group2-3, 8-24 years old), 'adult' (group4-5, 25-47 years old), 'elder' (group6-7, over 48 years old)}.

**Medical Disease Grading (MDG).** We select a series of Diabetic Retinopathy (DR) grading datasets, including Eyepacs Dugas et al., DeepDR Liu et al. (2022), and APTOS Karthik & Dane (2019), as datasets for medical disease grading. In these datasets, images are annotated in five levels of diabetic retinopathy from grade 1 to 5. We also add text-defined rating labels for all the levels: {'normal' (1), 'mild' (2), 'moderate' (3), 'severe' (4), 'extreme' (5)}. For coarse intermediate thoughts, the candidates are: {'normal' (1), 'early' (2-3), 'late' (4-5)}.

**Historical Date Estimation (HDE).** We select the HCI dataset Palermo et al. (2012) as the dataset for historical date estimation, which aims to estimate the decades of historical color photos. There are five decades, from 1930s to 1970s, annotated as 1 to 5. We add text-defined rating labels for each phase: {'early' (phase1, 1930s), 'early-mid' (phase2, 1940s), 'middle' (phase3, 1950s), 'mid-late' (phase4, 1960s), 'late' (phase5, 1970s)}. For coarse intermediate thoughts, the candidates are: {'before middle' (phase1-2, 1930s-1940s), 'middle' (phase3, 1950s), 'after middle' (phase4-5, 1960s-1970s)}.

## 4 ENHANCING MLLMs WITH ALL-IN-ONE VISUAL RATING CAPABILITIES

**Model Pipeline.** Fig. 3 presents an overview of the model pipeline, which mainly consists of three parts: Vision Encoder, Text Candidate Generation, and Coarse-to-fine CoT. The Vision Encoder processes visual input and encodes it into a series of visual tokens. Text Candidate Generation provides both coarse and fine text definitions, which will act as prompts and form a new question to instruct the LLM to provide an intermediate coarse thought for the coarse-to-fine CoT. Coarse-

Table 2: Accuracy performance of the visual rating benchmark (higher is better). "Tra." indicates the datasets used for fine-tuning. "Zero" denotes the model without fine-tuning. "Lite" denotes that the model is fine-tuned on the lite vision of datasets. "Full" denotes that the model is fine-tuned on the full datasets. *Datasets highlighted in gray indicate that their training splits are **not** used in our model's fine-tuning phase, which act as **zero-shot** testing.*

| MLLM | Tra. | IQA[1] | | | FAE | | | |
|---|---|---|---|---|---|---|---|---|
| | | SPAQ | ChallengeDB | KonIQ | Adience | CACD | Morph | UTK |
| LLaVA-1.5-7B | Zero | 0.243 | 0.296 | 0.396 | 0.452 | - | - | - |
| | Lite | 0.259 | 0.249 | 0.263 | 0.333 | - | - | - |
| Qwen2.5-VL-3B | Zero | 0.512 | **0.472** | 0.493 | 0.444 | - | - | - |
| | Lite | **0.600** | 0.446 | 0.561 | 0.480 | - | - | - |
| MiniGPT-v2-7B | Zero | 0.300 | 0.369 | 0.387 | 0.538 | - | - | - |
| | Lite | 0.298 | 0.369 | 0.488 | 0.426 | - | - | - |
| BLIP2-opt-2.7B | Zero | 0.144 | 0.185 | 0.158 | 0.087 | - | - | - |
| | Lite | 0.367 | 0.262 | 0.183 | 0.142 | - | - | - |
| InternVL-2B | Zero | 0.317 | 0.279 | 0.333 | 0.252 | - | - | - |
| | Lite | 0.352 | 0.348 | 0.372 | 0.198 | - | - | - |
| STORM-3B | Lite | 0.583 | 0.468 | **0.582** | 0.534 | - | - | - |
| | Full | 0.585 | 0.466 | 0.568 | **0.551** | - | - | - |

| MLLM | Tra. | IAA | | | MDG | | | HDE | Average |
|---|---|---|---|---|---|---|---|---|---|
| | | TAD66K | AVA | Aes. | Eyepacs | DeepDR | APTOS | HCI | |
| LLaVA-1.5-7B | Zero | 0.137 | 0.096 | 0.030 | 0.028 | 0.090 | 0.057 | 0.258 | 0.189 |
| | Lite | 0.354 | 0.591 | 0.583 | 0.547 | 0.248 | 0.445 | 0.220 | 0.372 |
| Qwen2.5-VL-3B | Zero | 0.207 | 0.275 | 0.081 | 0.073 | 0.158 | 0.191 | 0.265 | 0.288 |
| | Lite | 0.338 | 0.546 | 0.260 | 0.731 | 0.433 | 0.506 | 0.273 | 0.466 |
| MiniGPT-v2-7B | Zero | 0.197 | 0.179 | 0.047 | 0.164 | 0.218 | 0.273 | 0.250 | 0.266 |
| | Lite | 0.268 | 0.543 | 0.457 | 0.732 | 0.425 | 0.492 | 0.189 | 0.426 |
| BLIP2-opt-2.7B | Zero | 0.184 | 0.117 | 0.050 | 0.731 | 0.433 | 0.492 | 0.129 | 0.247 |
| | Lite | 0.273 | 0.188 | 0.196 | 0.714 | 0.433 | 0.495 | 0.187 | 0.313 |
| InternVL-2B | Zero | 0.284 | 0.348 | 0.286 | 0.254 | 0.240 | 0.309 | 0.204 | 0.282 |
| | Lite | 0.347 | 0.449 | 0.434 | 0.567 | 0.388 | 0.434 | 0.303 | 0.381 |
| STORM-3B | Lite | **0.370** | 0.650 | 0.658 | 0.734 | **0.435** | **0.508** | 0.341 | 0.533 |
| | Full | 0.368 | **0.655** | **0.668** | **0.741** | **0.435** | 0.506 | **0.424** | **0.542** |

to-fine CoT generates a finer final answer based on the coarse thought. Our STORM chooses Qwen2.5-VL-3B Bai et al. (2023b) as the LLM backbone. For more details, see Appendix B.

**Text Candidate Generation.** For different domain tasks, we first use GPT to generate a text definition for each numeric label. Then, manual adjustments are applied to make the text definition more realistic and compatible with human rating practices. After this, both the intermediate coarse thought and final answer have a text label and a numeric label. This offers several advantages: 1) Reducing digital hallucination. Since MLLMs are pre-trained using CLIP to align images and text rather than numbers, they are prone to numerical hallucination. By supplementing numeric labels with text definitions, MLLMs can learn more ordinal semantic relationships and reduce digital hallucination. 2) Differentiating task specificity. Since different tasks may share identical label ranges (e.g., 1-5 ratings) while having distinct task natures, the models could confuse label distributions across tasks. Leveraging textual definitions allows the models to capture task-specific specificity, while numeric labels can preserve the ordinal commonality essential for diverse rating tasks.

**Coarse-to-fine CoT.** To train an MLLM with our newly generated data, we add a CoT prompt (*"Please provide the coarse category that can help you answer the question better. The candidates is: "*), followed by text along with numeric category candidates for the question. The MLLM is instructed to perform the following three steps: (1) Make a coarse rating thought with the candidates (e.g. Child (0-10 years old), Teenager (11-20 years old), Youth (21-30 years old), Adult (31-44 years old),

---

[1]Here we use Accuracy and MAE for the IQA tasks to unify the metrics. For the commonly used metrics for IQA tasks, PLCC and SRCC, are provided in the supplementary materials, Appendix D.

Table 3: MAE performance of the visual rating benchmark (lower is better).

| MLLM | Tra. | IQA[1] | | | FAE | | | |
|---|---|---|---|---|---|---|---|---|
| | | SPAQ | ChallengeDB | KonIQ | Adience | CACD | Morph | UTK |
| LLaVA-1.5-7B | Zero | 1.294 | 1.155 | 0.852 | 0.859 | 11.439 | 9.251 | 11.763 |
| | Lite | 0.983 | 1.017 | 0.919 | 0.990 | 8.776 | 6.691 | 9.934 |
| Qwen2.5-VL-3B | Zero | 0.534 | **0.592** | 0.547 | 0.734 | 9.746 | **5.470** | 6.534 |
| | Lite | **0.423** | 0.605 | 0.469 | 0.715 | **7.541** | 7.589 | 6.433 |
| MiniGPT-v2-7B | Zero | 1.084 | 0.944 | 0.791 | 0.705 | 10.813 | 27.923 | 13.173 |
| | Lite | 1.143 | 0.922 | 0.638 | 0.919 | 10.075 | 9.511 | 11.837 |
| BLIP2-opt-2.7B | Zero | 1.846 | 1.721 | 1.681 | 3.734 | 10.436 | 14.156 | 28.921 |
| | Lite | 1.003 | 1.350 | 1.478 | 3.113 | 10.160 | 13.090 | 21.598 |
| InternVL-2B | Zero | 0.982 | 1.056 | 0.947 | 1.592 | 8.865 | 17.448 | 23.265 |
| | Lite | 0.902 | 0.939 | 0.876 | 1.980 | 8.683 | 10.776 | 15.695 |
| STORM-3B | Lite | 0.442 | 0.597 | **0.431** | 0.636 | 8.202 | 5.975 | 5.879 |
| | Full | 0.441 | 0.602 | 0.460 | **0.602** | 8.014 | 5.886 | **5.689** |

| MLLM | Tra. | IAA | | | MDG | | | HDE | Average |
|---|---|---|---|---|---|---|---|---|---|
| | | TAD66K | AVA | Aes. | Eyepacs | DeepDR | APTOS | HCI | |
| LLaVA-1.5-7B | Zero | 1.594 | 1.390 | 1.739 | 2.507 | 2.085 | 2.161 | 1.333 | 3.530 |
| | Lite | 0.776 | 0.433 | 0.466 | 0.864 | 1.295 | 1.984 | 1.318 | 2.531 |
| Qwen2.5-VL-3B | Zero | 1.301 | 0.857 | 1.337 | 1.645 | 1.348 | 1.221 | 1.159 | 2.358 |
| | Lite | 0.886 | 0.474 | 0.868 | 0.537 | 1.285 | 1.107 | 1.181 | 2.155 |
| MiniGPT-v2-7B | Zero | 1.373 | 1.028 | 1.477 | 1.536 | 1.239 | 1.240 | 1.311 | 4.617 |
| | Lite | 0.831 | 0.465 | 0.562 | 0.536 | 1.285 | 1.134 | 1.992 | 2.989 |
| BLIP2-opt-2.7B | Zero | 1.405 | 1.371 | 1.577 | 0.541 | 1.260 | 1.123 | 1.962 | 5.124 |
| | Lite | 1.115 | 1.014 | 1.047 | 0.578 | 1.200 | 1.107 | 1.658 | 4.251 |
| InternVL-2B | Zero | 1.049 | 0.815 | 0.956 | 1.123 | 1.200 | 1.141 | 1.364 | 4.407 |
| | Lite | 0.909 | 0.660 | 0.702 | 0.832 | **1.028** | **0.929** | 1.171 | 3.292 |
| STORM-3B | Lite | **0.726** | 0.363 | 0.360 | 0.511 | 1.280 | 1.098 | 0.924 | 1.958 |
| | Full | 0.730 | **0.354** | **0.351** | **0.495** | 1.280 | 1.106 | **0.689** | **1.907** |

Middle (45-60 years old), Elder (60+ years old)). (2) Based on the coarse rating thought, proceed to make a final answer. (3) Check that the coarse rating thought and answer are consistent. This strategy is designed to alleviate the problem of inconsistency between coarse intermediate thought and final answer, that is, to prevent the coarse intermediate thought from not including the final answer.

This methodology aims to serve three key objectives. 1) First and foremost, through a coarse-to-fine progressive analysis process, it allows to learn universal solutions for ordinal regression to endow the models with the all-in-one visual rating capability. This hierarchical approach is universally applicable to ordinal regression problems, as only their ordered categorical nature permits merging of adjacent categories for candidate reduction, enabling recursive hierarchical decomposition of the problem. 2) It transforms a multi-class rating problem into several smaller rating tasks with fewer candidate categories, therefore reducing the classification complexity through progressive candidate pruning. 3) Coarse labels are equivalent to merged neighboring categories, partially helping alleviate the class imbalance issues through category aggregation. For more VQA illustrations on other datasets, see Appendix E.

## 5 EXPERIMENTS

### 5.1 VISUAL RATING BENCHMARK

Our visual rating benchmark primarily focuses on scenarios where the MLLMs need to concentrate on ordinal understanding based on the visual input. Our experiments utilize 14 source datasets, and when an official training/evaluation split exists, we adopt it. In the cases where such a split does not exist, we randomly divide the dataset. Additionally, we incorporate the test splits of HCI, CACD, UTK, Aesthetic, KonIQ, and APTOS to evaluate the model's zero-shot visual rating capabilities.

Table 4: Ablation study on different instruct prompt strategies. "w/o CoT" denotes a standard, non-CoT-based inference process. "Only Num." and "Only Text" use only numeric and only text instruct prompts, respectively. "Num. + Text" uses both numeric and text instruct prompts.

| Instruct Prompt Strategy | Metric | IQA | | | FAE | | | |
| | | SPAQ | ChallengeDB | KonIQ | Adience | CACD | Morph | UTK |
| --- | --- | --- | --- | --- | --- | --- | --- | --- |
| w/o CoT | ACC | **0.600** | 0.446 | 0.561 | 0.480 | - | **-** | - |
| | MAE | **0.423** | 0.605 | 0.469 | 0.715 | **7.541** | 7.589 | 6.433 |
| Only Num. | ACC | 0.573 | 0.399 | 0.547 | 0.531 | - | - | - |
| | MAE | 0.461 | 0.751 | 0.487 | 0.674 | 9.856 | 9.620 | 9.464 |
| Only Text | ACC | 0.542 | 0.391 | 0.537 | 0.532 | - | - | - |
| | MAE | 0.495 | 0.717 | 0.503 | 0.665 | 9.412 | 9.326 | 8.298 |
| Num. + Text | ACC | 0.583 | **0.468** | **0.582** | **0.534** | - | - | - |
| | MAE | 0.442 | **0.597** | **0.431** | 0.636 | 8.202 | **5.975** | **5.879** |

| Instruct Prompt Strategy | Metric | IAA | | | MDG | | | HDE | Average |
| | | TAD66K | AVA | Aes. | Eyepacs | DeepDR | APTOS | HCI | |
| --- | --- | --- | --- | --- | --- | --- | --- | --- | --- |
| W/o CoT | ACC | 0.338 | 0.546 | 0.260 | 0.731 | 0.385 | 0.506 | 0.273 | 0.466 |
| | MAE | 0.886 | 0.474 | 0.868 | 0.537 | 1.348 | 1.107 | 1.181 | 2.155 |
| Only Num. | ACC | 0.351 | 0.622 | 0.364 | 0.716 | 0.433 | 0.504 | 0.326 | 0.487 |
| | MAE | 0.831 | 0.388 | 0.734 | 0.557 | 1.285 | 1.185 | **0.909** | 2.585 |
| Only Text | ACC | 0.351 | 0.609 | 0.434 | 0.731 | 0.433 | **0.514** | 0.265 | 0.485 |
| | MAE | 0.831 | 0.403 | 0.650 | 0.537 | 1.285 | **1.085** | 1.023 | 2.516 |
| Num. + Text | ACC | **0.370** | **0.650** | **0.658** | **0.734** | **0.435** | 0.508 | **0.341** | **0.533** |
| | MAE | **0.726** | **0.363** | **0.360** | **0.511** | **1.280** | 1.098 | 0.924 | **1.958** |

Table 5: Ablation study on different training strategies.

| Training Datasets | IQA | | FAE | | IAA | | MDG | | HDE | | Average | |
| | ACC | MAE | ACC | MAE | ACC | MAE | ACC | MAE | ACC | MAE | ACC | MAE |
| --- | --- | --- | --- | --- | --- | --- | --- | --- | --- | --- | --- | --- |
| Single | 0.523 | 0.521 | 0.532 | 6.976 | 0.444 | 0.770 | 0.557 | 0.972 | 0.318 | 0.985 | 0.492 | 2.548 |
| Full | **0.544** | **0.490** | **0.534** | **5.173** | **0.562** | **0.483** | **0.559** | **0.963** | **0.341** | **0.924** | **0.533** | **1.958** |

## 5.2 PERFORMANCE EVALUATION

We comprehensively evaluate STORM across various visual rating tasks to thoroughly assess our model's ordinal understanding ability. Tab. 2 and Tab. 3 report the accuracy and MAE performances of our STORM benchmark and popular MLLMs, including LLaVA-1.5-7B Liu et al. (2023a), Qwen2.5-VL-3B Bai et al. (2023b), MiniGPT-v2-7B Zhu et al. (2023), BLIP2-opt-2.7B Li et al. (2023a), and InternVL-2B Chen et al. (2024). We test other MLLMs only on the lite version of our datasets, and test our STORM on both the lite and full versions. By comparing the results of different models without fine-tuning and with fine-tuning on the lite version, we observe that after fine-tuning, the model significantly improves performances across all the datasets. This demonstrates the effectiveness of our proposed dataset. Notably, our STORM shows remarkable improvement in zero-shot performance when the training splits for the corresponding datasets are not utilized for model training. For instance, on the Aes. datasets, our model achieves nearly $2.5\times$ performance compared to the Qwen2.5-VL pipeline without a coarse-to-fine CoT process. Furthermore, the STORM pipeline trained on the lite versions yields superior results on the HCI task which is a zero-shot domain not appearing during the training process, showing the efficacy of our benchmark in enhancing the model's universal visual rating abilities. The STORM pipeline trained on the full versions achieves the best performances on both in-domain and out-of-domain tasks, which validate the effectiveness and potential of our dataset.

## 5.3 ABLATION STUDIES

In the ablation studies below, by default, we ablate STORM-3B that is trained and evaluated on the lite version of our datasets with the proposed coarse-to-fine CoT benchmark.

**Different Instruct Prompt Strategies.** Tab. 4 shows the performances of our model on the lite version of the visual rating benchmark using different strategies for instruct prompts. As anticipated, the model not employing coarse-to-fine CoT yields lower performance, which indicates inherent challenges in directly predicting ratings. In contrast, our baseline with coarse-to-fine CoT performs better, especially on zero-shot datasets, illustrating the effectiveness of the coarse-to-fine CoT in

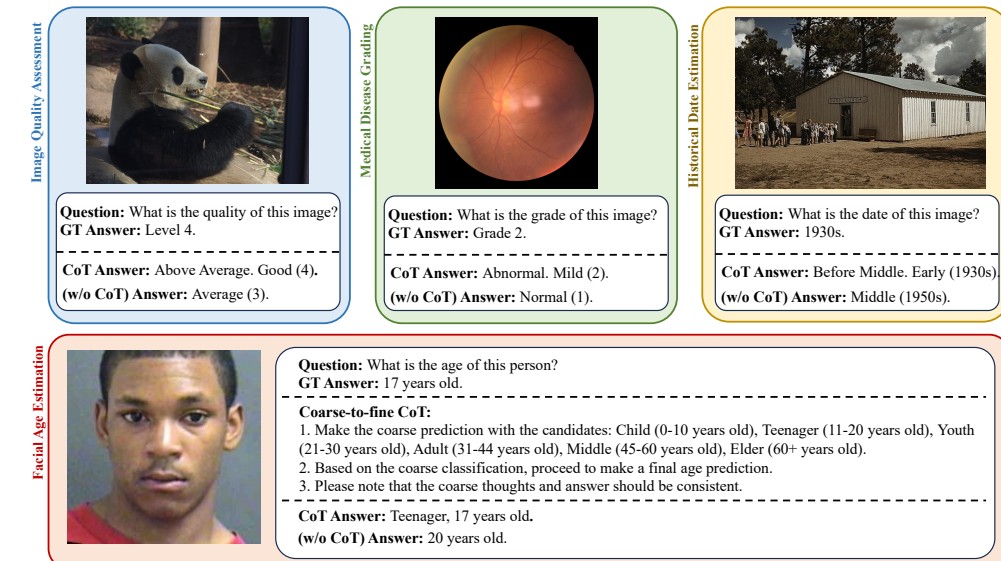

Figure 4: Visualization results of coarse-to-fine CoT on different datasets.

enhancing robust and general thinking ability for visual rating by learning the ordinal regression nature. In addition, compared to using only numeric labels or text definitions, the MLLM with both numeric labels and text definitions achieves the best performance, showing the effect of both digital and semantic instructions. Notably, text proves to be more effective than numbers, which validates our previous hypothesis that LLMs pre-trained with CLIP are more sensitive to text prompts.

**Different Training Strategies.** We conduct ablation experiments on different selections of training data. For each domain task, we compare the model's performances after being trained on single-domain datasets versus being trained on all domain datasets. The results are shown in the top part of Table 5, which indicate that the model performs better after training on all the domains compared to training only on a single domain. This demonstrates that the model can learn generalized and useful ordinal regression properties from different domain tasks, therefore improving the overall performance across various visual rating tasks. It also highlights the advantages and effectiveness of our benchmark and datasets. We also investigate the LLM performance under different fine-tuning methods, e.g, Low-Rank Adaptation (LoRA) Hu et al. (2022) and Full Fine-Tuning, see Appendix D.

### 5.4 VISUALIZATION

We visually display STORM's performance qualitatively in Fig. 4, highlighting its visual rating ability to conduct a coarse-to-fine CoT process and provide trustworthy predictions. Despite variations in label definitions and ranges across different tasks, the inherent commonality in ordinal nature of labels enables a unified thinking paradigm through progressive refinement of label granularity, achieving coarse-to-fine estimation across these visual rating tasks.

## 6 CONCLUSIONS

In this paper, we introduced STORM, a pioneering approach that enhances multi-modal large language models with the all-in-one visual rating capability. This methodology addresses critical gaps in MLLMs, especially in interpretability and processing of dynamic visual input. Our STORM data collection offers 655K annotated question-answer pairs from diverse ordinal regression tasks for comprehensive visual rating learning. Our novel coarse-to-fine processing pipeline allows MLLMs to learn a universal paradigm of ordinal regression and provide intermediate interpretable thoughts. STORM offers a general and trustworthy paradigm for tackling diverse visual rating tasks, and our visual rating benchmark advances the evaluation of MLLMs on both in-domain and out-of-domain tasks. Extensive experiments validated the framework's effectiveness and robustness, putting forward a promising basis for further exploration in visual rating.

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

# STORM: Benchmarking Visual Rating of MLLMs with a Comprehensive Ordinal Regression Dataset

## Supplementary Material / Appendix

## A    APPENDIX OVERVIEW

Our supplementary includes the following sections:

- **Section B: Framework details.** Details for model design, implementation and training data.
- **Section C: More Dataset Details and Visualization.** More Details and Visualization of our dataset and demos.
- **Section D: More experiment results.** Additional performance evaluation and performance analysis.
- **Section E: Prompt design.** Prompt for generating the coarse-to-fine CoT dataset and evaluating the performance.
- **Section F: Limitations.** Discussion of limitations of our work.
- **Section G: Potential negative societal impacts.** Discussion of potential negative societal impacts of our work.
- **Section H: Disclaimer.** Disclaimer for the visual rating dataset and the related model.
- **Section I: Use of LLM.** Describe the usage of LLM.

**Reproducibility Statement:** We offer the anonymous Datasets and code links below to ensure our framework can be reproduced easily.

| Artifcat | Link | License |
|---|---|---|
| Code Repository | `https://anonymous.4open.science/r/STORM-CDC7/README.md` | Apache-2.0 license |
| Data | `https://huggingface.co/datasets/ttlyy/ORD` | CC BY 4.0 |
| Model Weights | `https://huggingface.co/datasets/ttlyy/ORD` | Apache-2.0 license |

The authors are committed to ensuring its regular updates.

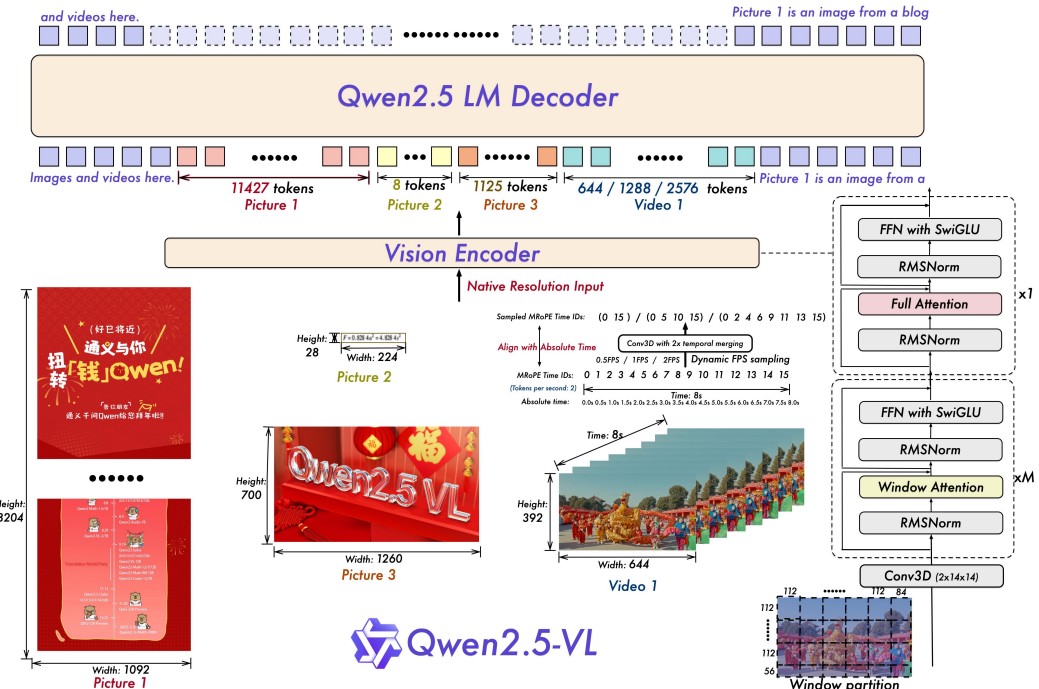

Figure 5: Overview of Qwen-2.5-VL pipeline.

# B FRAMEWORK DETAILS

## B.1 MODEL DETAILS

For LLaVA-1.5-7B, we choose the pre-trained ViT-L/14 of CLIP Radford et al. (2021b) as the vision encoder and Vicuna-7B Chiang et al. (2023) as our LLM, which has better instruction following capabilities in language tasks compared to LLaMA Touvron et al. (2023). For Qwen2.5-VL-3B, the vision encoder the native dynamic resolution ViT. The overview of Qwen-2.5-VL Bai et al. (2023b) are shown in Fig. 5. Considering an input original image, we take the vision encoder to obtain the visual feature. Our STORM-3B employes Qwen-2.5-VL-3B as the backbone.

## B.2 IMPLEMENTATION DETAILS

Our model undergoes a two-stage training process. In the first stage, we pre-train the model for 1 epoch using a learning rate of 2e-3 and a batch size of 128. For the second stage, we fine-tune the model for 1 epoch on our visual rating dataset, employing a learning rate of 2e-5 and a batch size of 128. The Adam optimizer with zero weight decay and a cosine learning rate scheduler are utilized. To conserve GPU memory during fine-tuning, we employ FSDP (Full Shard Data Parallel) with ZeRO3-style. All models are trained using $32 \times$ A100s. In the case of training the setting with a 7B LLM and a resolution of 224, the first/second pre-training stage completes within 1/16 hours.

# C MORE DATASET DETAILS AND VISUALIZATION

## C.1 DATASETS TRAINING AND TESTING SPLIT.

In this section, we provide the sample numbers of training and test split of all datasets, as shown in Tab. 6 and Tab. 7.

Table 6: Training and testing split of IQA and IAA domain datasets. Training split includes full version and lite version.

| Dataset | SPAQ | CDB | KonIQ | AVA | TAD66K | Aesthetic |
|---|---|---|---|---|---|---|
| Training Full | 8900 | 936 | - | 229958 | 52224 | - |
| Training Lite | 8900 | 936 | - | 25551 | 13056 | - |
| Testing | 2225 | 233 | 2014 | 25550 | 14076 | 1370 |

Table 7: Training and testing split of FAE, MDG and HDE domain datasets. Training split includes full version and lite version.

| Dataset | Adience | CACD | Morph | UTK | Eyepacs | DeepDR | APTOS | HCI |
|---|---|---|---|---|---|---|---|---|
| Training Full | 15589 | 147102 | 40012 | - | 31599 | 1200 | - | - |
| Training Lite | 15589 | 16345 | 10003 | - | 31599 | 1200 | - | - |
| Testing | 1732 | 16344 | 10003 | 2410 | 3527 | 400 | 366 | 132 |

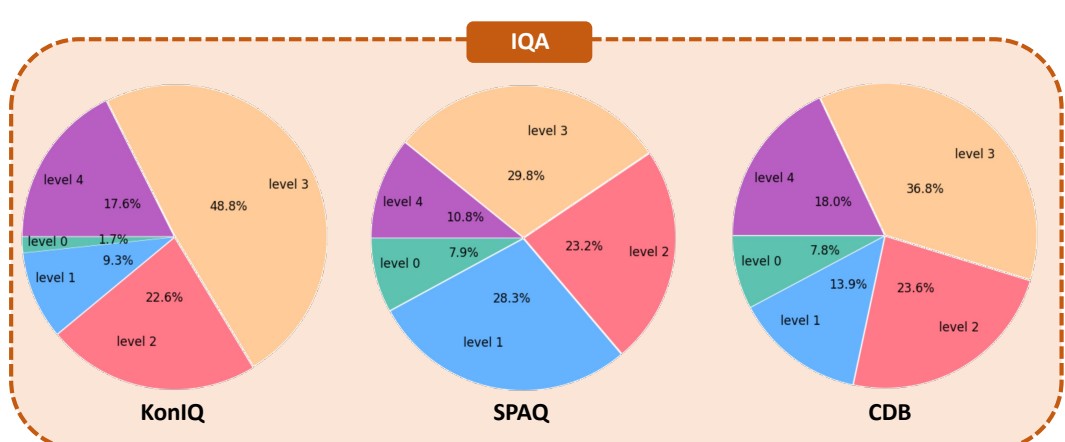

Figure 6: Statistics of the IQA domain datasets.

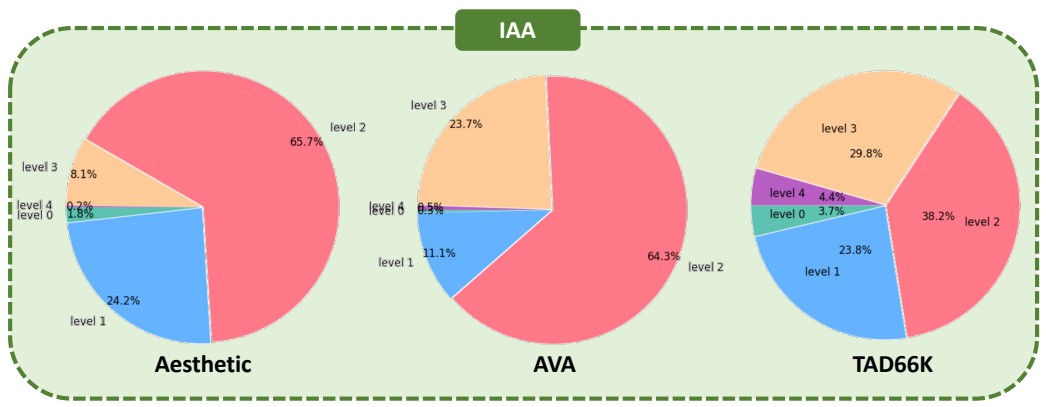

Figure 7: Statistics of the IAA domain datasets.

## C.2 DATASETS DISTRIBUTION VISUALIZATION.

In this section, we provide a visualization of the data statistics. We partition the category distribution of each dataset in Fig. 6, Fig. 7, Fig. 8, Fig. 9.

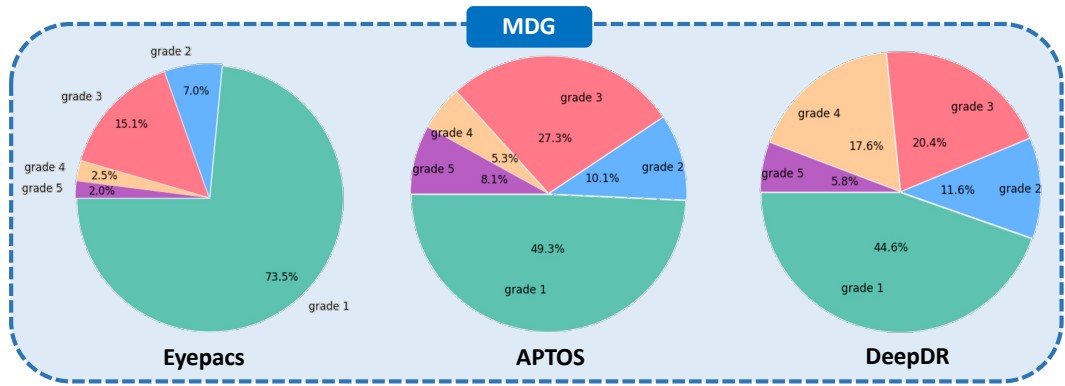

Figure 8: Statistics of the MDG domain datasets.

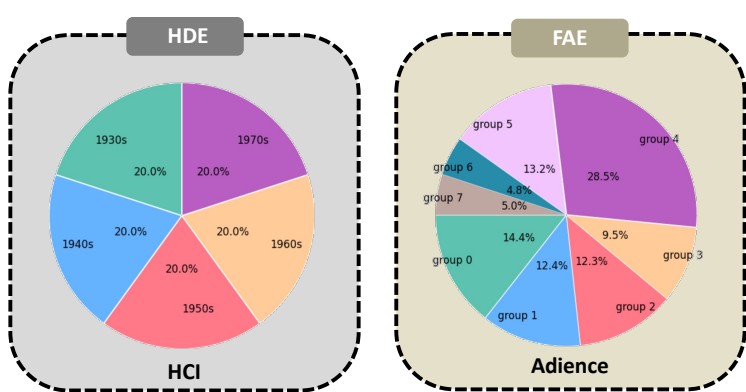

Figure 9: Statistics of the FAE and HDE domain datasets.

# D MORE EXPERIMENT RESULTS

## D.1 LARGER STORM MODEL

Tab. 8 and Tab. 8 show the performance of STORM-7B using Qwen2.5-VL-7B as the backbone. However, the performance is not much different from the 3B version. Therefore, we choose STORM-3B as the final model.

## D.2 PLCC AND SRCC PERFORMANCE IN IQA TASKS.

Tab. 10 show the SRCC and PLCC results. It can be seen that our STORM achieves the best performance both in SRCC and PLCC on all IQA datasets, showing the effectiveness of our method.

## D.3 DIFFERENT FINE-TUNING STRATEGIES.

To explore the effect of different parameter fine-tuning methods for LLMs. We compare the commonly used Low-Rank Adaptation (LoRA) Hu et al. (2022) and Full Fine-Tuning (FFT) methods, and report the results in the lower part of Tab. 11. One can observe that FTT performs better and is more robust. Hence, we adopt FTT for all the fine-tuning experiments.

## D.4 CONFUSION MATRIXES ANALYSIS

We provide more visualization results of confusion matrixes of our STORM on zero-shot datasets in Fig. 10, Fig. 11, Fig. 12, Fig. 13 and Fig. 14.

Table 8: ACC performance of the STORM-7B.

| MLLM | Tra. | IQA | | | FAE | | | |
|---|---|---|---|---|---|---|---|---|
| | | SPAQ | ChallengeDB | KonIQ | Adience | CACD | Morph | UTK |
| STORM-3B | Lite | **0.583** | **0.468** | **0.582** | **0.534** | - | - | - |
| STORM-7B | Lite | 0.514 | 0.438 | 0.543 | 0.503 | - | - | - |

| MLLM | Tra. | IAA | | | MDG | | | HDE | Average |
|---|---|---|---|---|---|---|---|---|---|
| | | TAD66K | AVA | Aes. | Eyepacs | DeepDR | APTOS | HCI | |
| STORM-3B | Lite | **0.370** | 0.650 | **0.658** | **0.734** | **0.435** | **0.508** | **0.341** | 0.533 |
| STORM-7B | Lite | 0.367 | **0.654** | 0.541 | 0.177 | 0.340 | 0.429 | 0.250 | **0.432** |

Table 9: MAE performance of the STORM-7B.

| MLLM | Tra. | IQA | | | FAE | | | |
|---|---|---|---|---|---|---|---|---|
| | | SPAQ | ChallengeDB | KonIQ | Adience | CACD | Morph | UTK |
| STORM-3B | Lite | **0.442** | **0.597** | **0.431** | **0.636** | 8.202 | 5.975 | 5.879 |
| STORM-7B | Lite | 0.562 | 0.652 | 0.496 | 0.641 | **7.776** | **5.405** | **5.508** |

| MLLM | Tra. | IAA | | | MDG | | | HDE | Average |
|---|---|---|---|---|---|---|---|---|---|
| | | TAD66K | AVA | Aes. | Eyepacs | DeepDR | APTOS | HCI | |
| STORM-3B | Lite | **0.726** | 0.363 | **0.360** | **0.511** | 1.280 | 1.098 | **0.924** | 1.958 |
| STORM-7B | Lite | 0.739 | **0.353** | 0.514 | 1.481 | **1.093** | **1.003** | 1.129 | **1.953** |

Table 10: SRCC and PLCC results of all models in IQA tasks.

| MLLM | Tri. | SPAQ | | CDB | | KonIQ | |
|---|---|---|---|---|---|---|---|
| | | PLCC | SRCC | PLCC | SRCC | PLCC | SRCC |
| LLaVA-1.5-7B | Zero | -0.034 | -0.007 | 0.094 | 0.130 | -0.007 | -0.014 |
| LLaVA-1.5-7B | Lite | 0.037 | 0.036 | -0.008 | -0.005 | 0.008 | 0.009 |
| MiniGPT-v2-7B | Zero | 0.384 | 0.376 | 0.192 | 0.198 | 0.279 | 0.266 |
| BLIP2-opt-2.7B | Zero | -0.067 | -0.115 | -0.085 | -0.073 | -0.070 | -0.077 |
| BLIP2-opt-2.7B | Lite | 0.172 | 0.173 | 0.101 | 0.090 | 0.038 | 0.049 |
| InternVL-2B | Zero | 0.333 | 0.327 | 0.134 | 0.134 | 0.095 | 0.088 |
| InternVL-2B | Lite | 0.365 | 0.354 | 0.229 | 0.233 | 0.107 | 0.110 |
| Qwen2.5-VL-3B | Zero | 0.778 | 0.790 | 0.690 | 0.683 | 0.597 | 0.531 |
| Qwen2.5-VL-3B | Lite | 0.787 | 0.803 | 0.575 | 0.537 | 0.677 | 0.627 |
| STORM-3B | Lite | 0.804 | 0.817 | 0.675 | 0.656 | 0.692 | 0.650 |
| STORM-3B | Full | **0.808** | **0.822** | **0.686** | **0.674** | **0.727** | **0.690** |

Table 11: Ablation study on different training strategies.

| Fine-tuning Strategy | IQA | | FAE | | IAA | | MDG | | HDE | | Average | |
|---|---|---|---|---|---|---|---|---|---|---|---|---|
| | ACC | MAE | ACC | MAE | ACC | MAE | ACC | MAE | ACC | MAE | ACC | MAE |
| LoRA Hu et al. (2022) | 0.171 | 1.522 | 0.189 | 8.301 | 0.199 | 1.041 | 0.553 | 0.985 | 0.227 | 1.311 | 0.289 | 3.225 |
| FFT | **0.544** | **0.490** | **0.534** | **5.173** | **0.562** | **0.483** | **0.559** | **0.963** | **0.341** | **0.924** | **0.533** | **1.958** |

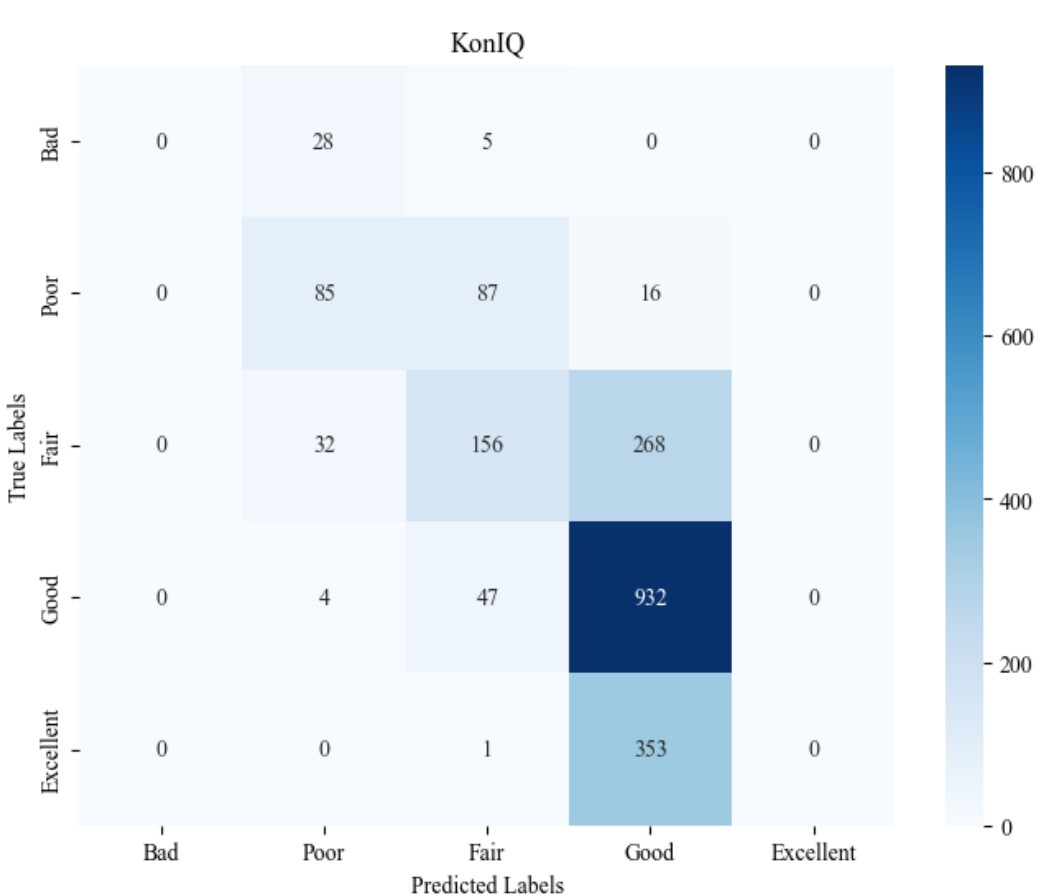

Figure 10: Confusion matrixes visualization results of the STORM on the KonIQ dataset.

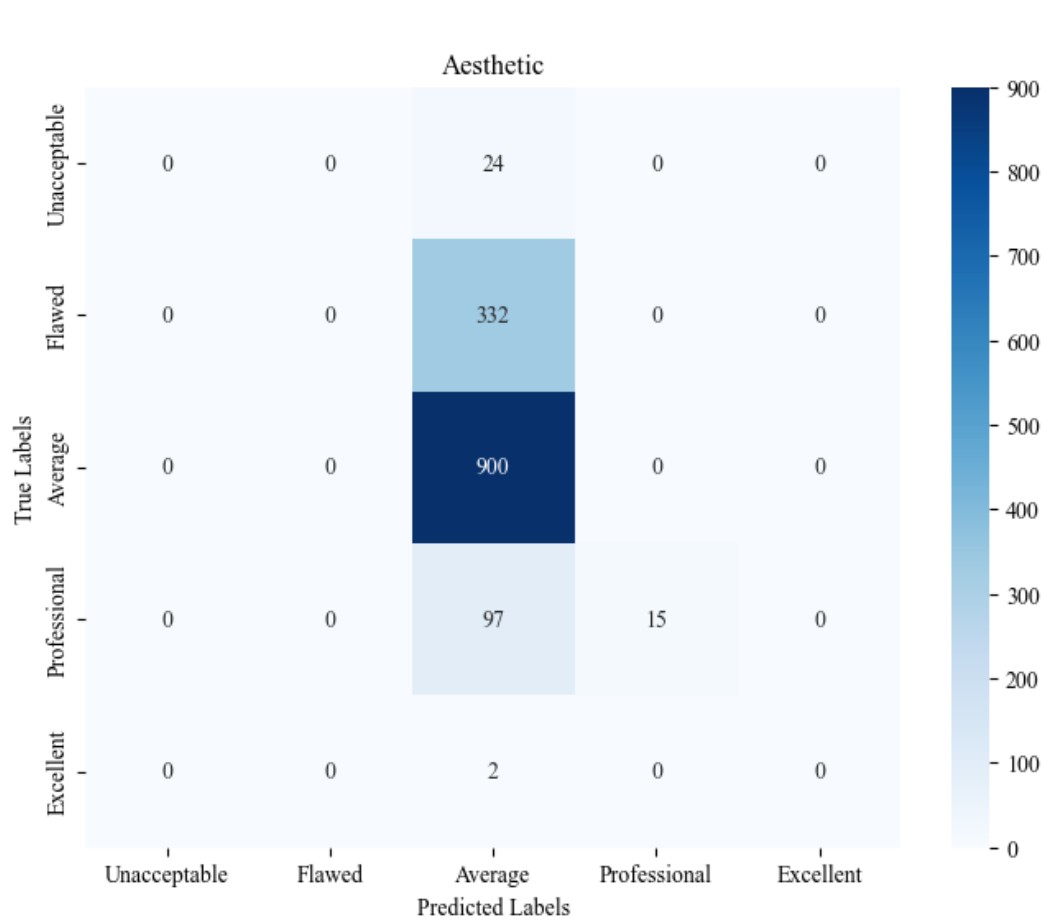

Figure 11: Confusion matrixes visualization results of the STORM on the Aesthetic dataset.

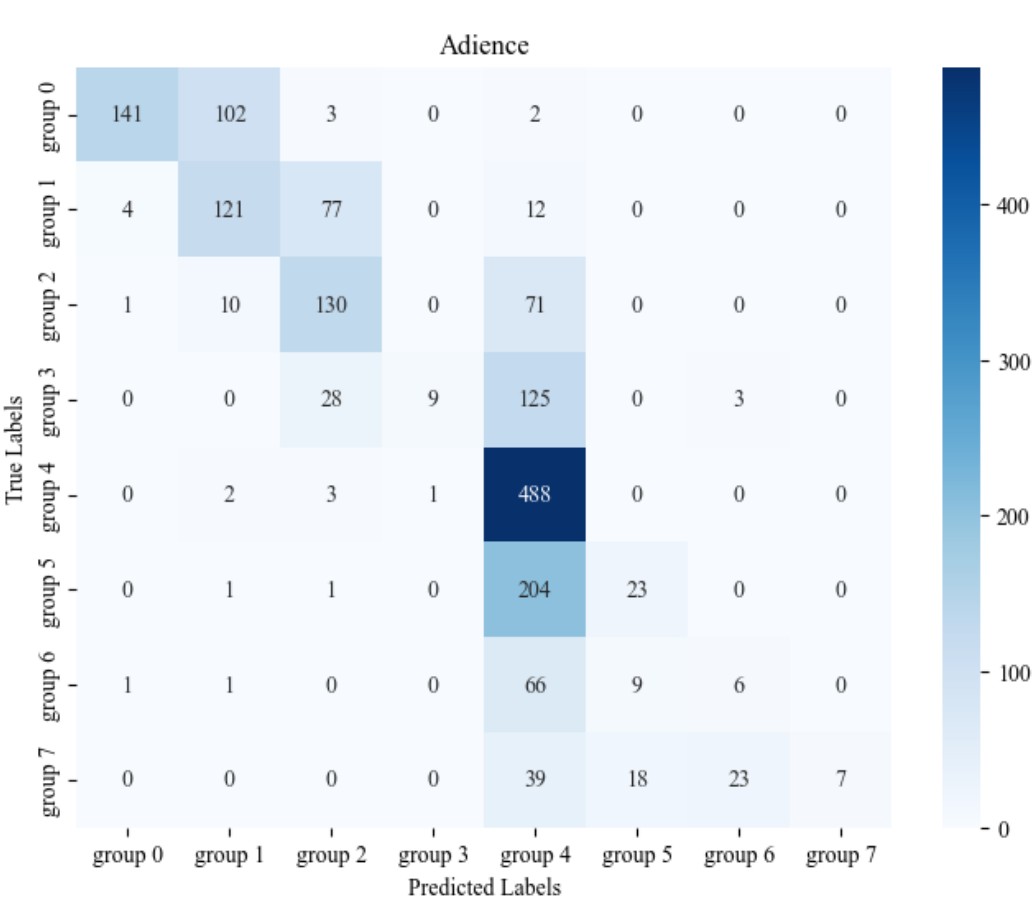

Figure 12: Confusion matrixes visualization results of the STORM on the Adience dataset.

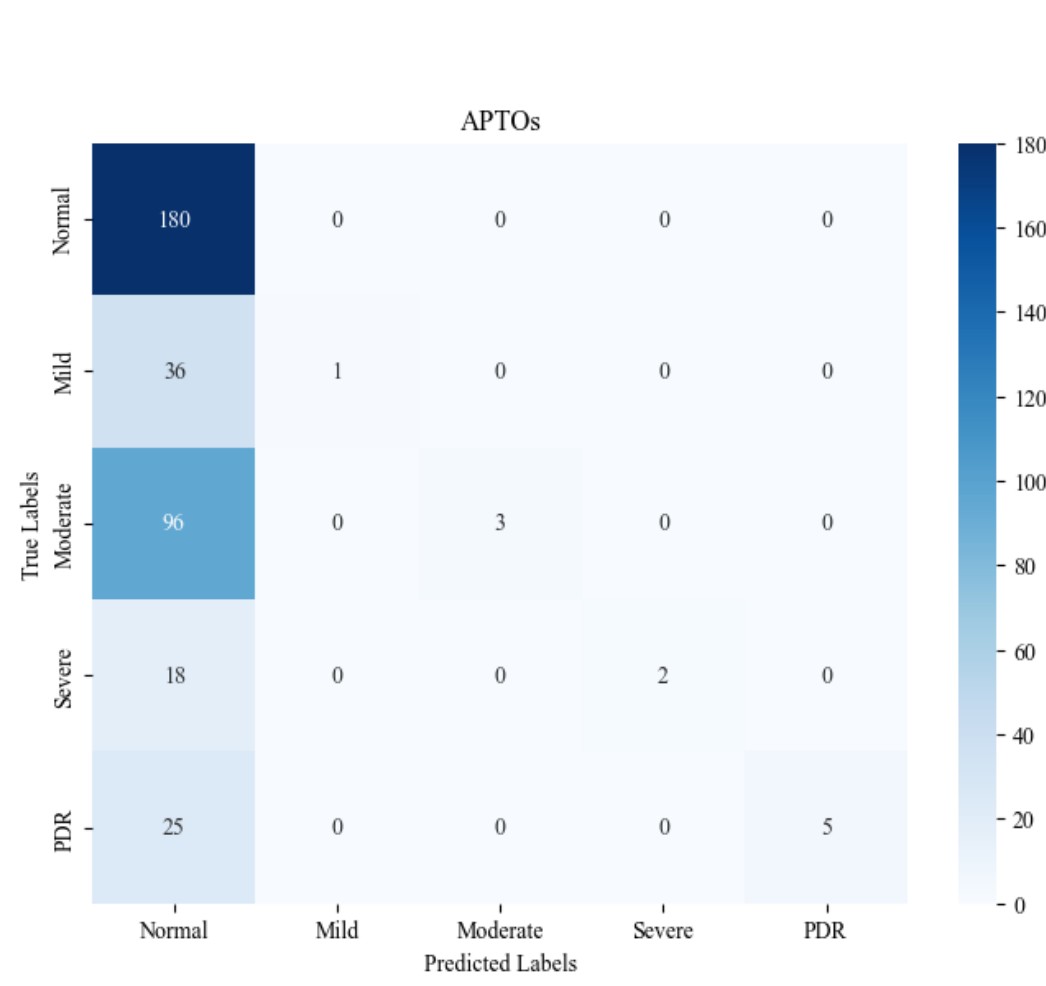

Figure 13: Confusion matrixes visualization results of the STORM on the APTOS dataset.

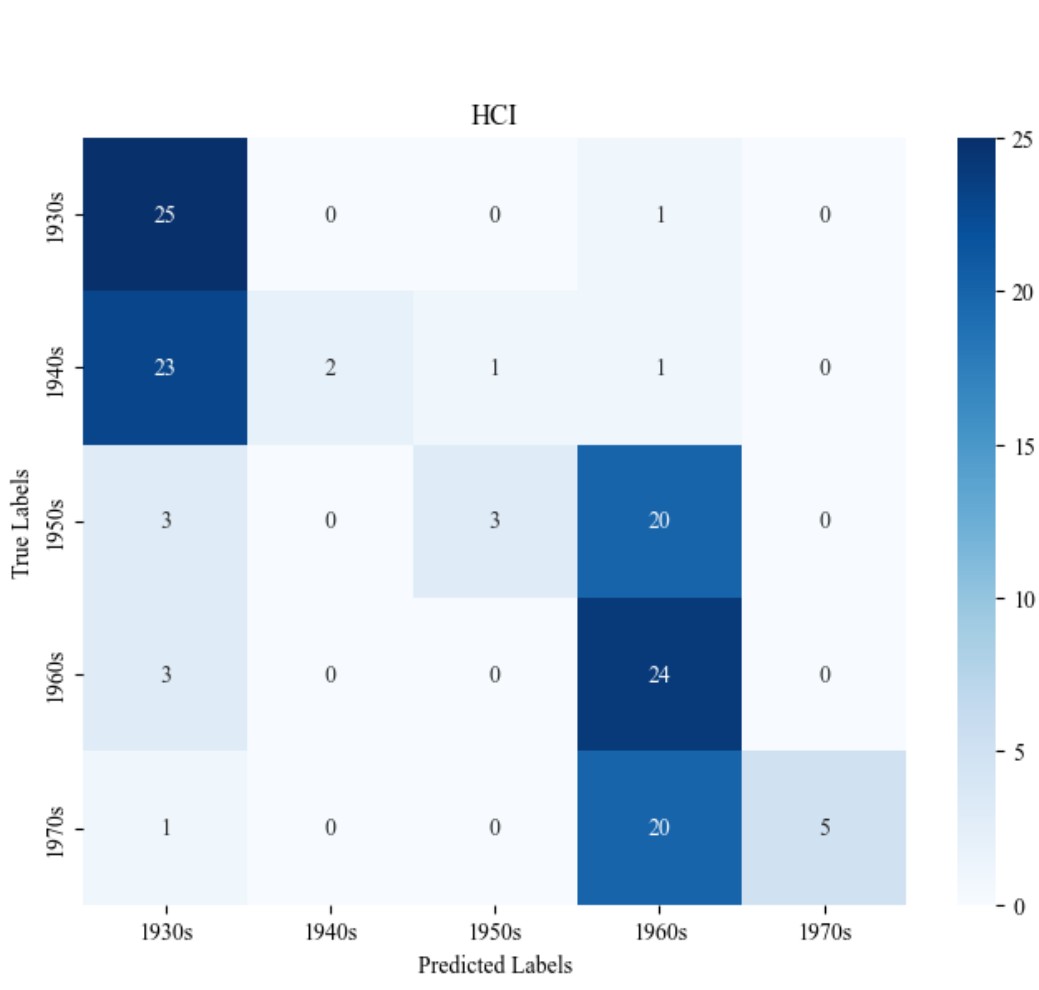

Figure 14: Confusion matrixes visualization results of the STORM on the HCI dataset.

# E  PROMPT DESIGN

## E.1  GENERATING THE DATASET FOR IQA

> ¡image¿ You are now an advanced Image Quality Evaluator, and your task is to assess the quality of the provided image. Please evaluate the image's quality based on a 5-rate scale: rate0(Bad), rate1(Poor), rate2(Fair), rate3(Good), rate4(Excellent). Please provide the coarse category that can help you answer the question better. Please first coarsely categorise the image: rate0-1(Below Fair), rate2(Fair), rate3-4(Above Fair). Based on the coarse classification, proceed to make a final rate prediction. The specific steps are as follows:
>
> 1. Make the coarse prediction with the candidates:rate0-1(Below Fair), rate2(Fair), rate3-4(Above Fair).
>
> 2. Based on the coarse classification, proceed to make a final age prediction with the candidates: rate0(Bad), rate1(Poor), rate2(Fair), rate3(Good), rate4(Excellent).
>
> 3. Please note that the coarse thoughts and the final answer should be consistent.
>
> Answer: [Coarse answer], [Final answer]

## E.2  GENERATING THE DATASET FOR IAA

> ¡image¿ You are now an advanced Aesthetic Evaluation Evaluator, and your task is to assess the aesthetic quality of the provided image. Please evaluate the image's aesthetic quality based on a 5-level scale: level0(Unacceptable), level1(Flawed), level2(Average), level3(Professional), level4(Excellent). Please first coarsely categorise the image: level0-1(Below Average), level2(Average), level3-4(Above Average). Based on the coarse classification, proceed to make a final level prediction. The specific steps are as follows:
>
> 1. Make the coarse prediction with the candidates:level0-1(Below Average), level2(Average), level3-4(Above Average).
>
> 2. Based on the coarse classification, proceed to make a final age prediction with the candidates: level0(Unacceptable), level1(Flawed), level2(Average), level3(Professional), level4(Excellent).
>
> 3. Please note that the coarse thoughts and the final answer should be consistent.
>
> Answer: [Coarse answer], [Final answer]

## E.3  GENERATING THE DATASET FOR FAE

> ¡image¿ You are an experienced facial analysis expert, and you need to estimate the age group of the person in the provided facial image based on their facial features. The known age range of the image is from 16 to 77 years old. Please first coarsely categorise the image: Teenager(16-24 years old), Adult(25-47 years old), Elder(48+ years old). Based on the coarse classification, proceed to make a final age prediction.The final output should be in the format: Coarse Answer: [result], Predicted Age: [result]. The specific steps are as follows:
>
> 1. Make the coarse prediction with the candidates: Teenager(16-24 years old), Adult(25-47 years old), Elder(48+ years old).
>
> 2. Based on the coarse classification, proceed to make a final age prediction with the candidates: from 16 to 77 years old.
>
> 3. Please note that the coarse thoughts and the final answer should be consistent.
>
> Answer: Coarse answer], [Predicted Age]

### E.4 GENERATING THE DATASET FOR MDG

> ¡image¿ You are an experienced ophthalmologist, and you need to perform disease grading on the provided fundus image. These are all the candidate stages: stage0(no retinopathy), stage1(mild NPDR), stage2(moderate NPDR), stage3(severe NPDR) and stage4(PDR). Please first coarsely categorise the fundus: Normal(stage0), Early(stage1-2), Late(stage3-4). Based on the coarse classification, proceed to make a final stage prediction. The specific steps are as follows:
>
> 1. Make the coarse prediction with the candidates:Normal(stage0), Early(stage1-2), Late(stage3-4).
>
> 2. Based on the coarse classification, proceed to make a final age prediction with the candidates: stage0(no retinopathy), stage1(mild NPDR), stage2(moderate NPDR), stage3(severe NPDR) and stage4(PDR).
>
> 3. Please note that the coarse thoughts and the final answer should be consistent.
>
> Answer: [Coarse answer], [Predicted grade]

### E.5 GENERATING THE DATASET FOR HDE

> ¡image¿ You are now an advanced history researcher, and you need to grade the provided images by decade. These are all candidate categories: phase0(1930s), phase1(1940s), phase2(1950s), phase3(1960s), and phase4(1970s). Please first coarsely categorise the image: Early(phase0-phase1), Mid(phase2), Late(phase3-phase4). Based on the coarse classification, proceed to make a final phase prediction.The final output should be in the format: Coarse Classification: [result], Predicted Phase: [result]. The specific steps are as follows:
>
> 1. Make the coarse prediction with the candidates: Early(phase0-phase1), Mid(phase2), Late(phase3-phase4).
>
> 2. Based on the coarse classification, proceed to make a final age prediction with the candidates: phase0(1930s), phase1(1940s), phase2(1950s), phase3(1960s), and phase4(1970s).
>
> 3. Please note that the coarse thoughts and the final answer should be consistent.
>
> Answer: [Coarse answer], [Predicted Phase]

## F LIMITATIONS

The definitions of labels for different domain tasks are quite diverse.

In scenarios where the definitions of labels for different domain tasks are quite diverse, STORM may struggle to possess fluctuation in performance according to different text definitions generated of labels. This places a relatively high demand on the user's ability to accurately define corresponding text prompts of rating categories.

Our data pipeline inherits the limitations of utilizing GPT-4 API to generate text definition. (1) Accuracy and Misinformation: Generated content may not always be accurate, which could lead to the spread of misinformation. To mitigate this, we have designed a manual adjustment script as a post-process to improve text prompt quality. (2) Bias and Fairness: Since we do not have access to the training data of GPT-4, the generated instructional data might reflect inherent biases, potentially reinforcing social or cultural inequalities present in the base model training. In terms of data usage, we explicitly state that OpenAI's terms must be adhered to, and the data can only be used for research purposes.

## G POTENTIAL NEGATIVE SOCIETAL IMPACTS

The potential negative societal impacts of our work are similar to other MLLMs and LLMs. The development of CoT and MLLMs, while advancing AI, poses societal risks like increased privacy invasion, the perpetuation of biases, the potential for misinformation, job displacement, and ethical concerns regarding accountability and consent.

## H  DISCLAIMER

This dataset was collected and released solely for research purposes, with the goal of making the MLLMs dynamically focus on visual inputs and provide intermediate interpretable thoughts. The authors are strongly against any potential harmful use of the data or technology to any party.

**Intended Use.** The data, code, and model checkpoints are intended to be used solely for (I) future research on visual-language processing and (II) reproducibility of the experimental results reported in the reference paper. The data, code, and model checkpoints are not intended to be used in clinical care or for any clinical decision making purposes.

**Primary Intended Use.** The primary intended use is to support AI researchers reproducing and building on top of this work. STORM and its associated models should be helpful for exploring various vision question answering (VQA) research questions.

**Out-of-Scope Use.** Any deployed use case of the model — commercial or otherwise — is out of scope. Although we evaluated the models using a broad set of publicly-available research benchmarks, the models and evaluations are intended for research use only and not intended for deployed use cases.

## I  USE OF LLM

We employed LLMs solely for language polishing and manuscript refinement purposes. The LLM assistance was restricted to improving grammatical accuracy, sentence flow, and overall presentation clarity. All research content, methodology, analysis, and scientific conclusions were developed independently by the authors without LLM contribution. The LLM was not utilized for idea generation, experimental design, data interpretation, or scientific reasoning.

