# OpenReview forum: "STORM: Benchmarking Visual Rating of MLLMs with a Comprehensive Ordinal Regression Dataset"
_ICLR.cc/2026/Conference — ICLR 2026 Conference Withdrawn Submission_

### Official Review · Reviewer_K22g · 2025-10-26

**Soundness:** 2
**Presentation:** 2
**Contribution:** 3
**Rating:** 6
**Confidence:** 4

**Summary:**

This paper introduces the STORM benchmark, designed to evaluate the visual ordinal regression capabilities of multimodal large language models (MLLMs). The authors construct a comprehensive dataset comprising 14 datasets across 5 domains (image quality assessment, facial age estimation, medical disease grading, etc.), totaling 655K samples (Table 1). The core method is a coarse-to-fine Chain-of-Thought (CoT) pipeline (Figures 2-3), which first prompts the model to predict a coarse-grained category (e.g., “Youth”), and then, based on this, predicts a fine-grained result (e.g., “27 years old”). A mixed “text + number” label format is adopted to mitigate numerical hallucinations. Experimental results (Tables 2-3) show that the STORM-3B model trained on Qwen2.5-VL-3B outperforms baselines such as LLaVA-1.5 and MiniGPT-v2 in terms of average accuracy and MAE, and ablation studies (Table 4) validate the effectiveness of both the CoT and the mixed-label strategies.

**Strengths:**

- The primary contribution lies in providing the first large-scale, multi-domain benchmark for assessing MLLMs’ visual ordinal regression capabilities. The dataset construction is rigorous, covering 655K samples across five distinct domains (Table 1), filling a gap in this line of research.
- The experimental design is relatively comprehensive, comparing against multiple baseline models (Tables 2-3), and the detailed ablations (Table 4) substantiate the effectiveness of key design choices, especially the advantage of “Num. + Text” labels over using numeric-only or text-only labels.
- Cross-domain training experiments (Table 5) preliminarily demonstrate that joint training on multiple domains yields performance gains over single-domain training, providing empirical support for building universal visual rating models.

**Weaknesses:**

- The paper exhibits two fundamental issues. First, the so-called “coarse-to-fine CoT” is essentially closer to hierarchical supervision rather than genuine chain-of-thought reasoning. As indicated by Figure 2 and the description on page 5, lines 269-277, the model is trained to output both coarse and fine labels, but the paper does not clearly specify whether, at inference time, there exists a true dependency from the coarse prediction to the fine prediction. The setup of the “w/o CoT” baseline in Table 4 is also unclear, as the paper does not state whether this baseline uses the same “Num. + Text” labels, making it impossible to disentangle whether performance gains derive from the CoT structure itself or merely from the richer label format.
- Second, the argument for “universality” and “zero-shot” capability appears to involve circular reasoning. On page 7, lines 377-381, the paper claims “zero-shot” performance on datasets such as KonIQ, CACD, and APTOS, but these datasets actually belong to domains already used during training (IQA, FAE, MDG; see Table 1). They are merely different datasets within the same domains, which demonstrates cross-dataset generalization within a domain rather than true cross-domain zero-shot capability.

**Questions:**

Regarding the paper's methods and conclusions, I have the following questions:
1. What is the specific mechanism of "Coarse-to-fine CoT" during the inference phase? Does the model first generate the coarse thought and then generate the fine-grained answer conditioned on it, or does it jointly output both results? If it is the latter, why is it termed a "Chain-of-Thought" rather than "hierarchical supervision"?
2. In the ablation study in Table 4, how is the "w/o CoT" baseline implemented? Was it also trained using the "Num. + Text" labels for direct prediction? Can you provide a fair comparison to disentangle the respective gains from the CoT structure versus the rich label format?
3. The paper emphasizes "universal" rating capability, yet the "zero-shot" testing is still confined to the five major domains used during training. Can you provide test results on ordinal regression tasks in entirely new (out-of-domain) domains?
4. In Table 2 and Table 3, the performance of STORM-3B (Lite) on several tasks (e.g., IQA-KonIQ, FAE-Morph, FAE-UTK) is significantly worse than the un-fine-tuned Qwen2.5-VL-3B (Zero) baseline. Does this indicate that fine-tuning on the STORM Lite dataset caused negative transfer or catastrophic forgetting? How do the authors explain this phenomenon?

---

### Official Review · Reviewer_LFr8 · 2025-10-29

**Soundness:** 2
**Presentation:** 3
**Contribution:** 2
**Rating:** 2
**Confidence:** 4

**Summary:**

The manuscript introduces STORM, a large-scale data collection and benchmark designed to evaluate the Stimulating Trustworthy Ordinal Regression ability of MLLMs for general-purpose visual quality/rating tasks. STORM aggregates 14 ordinal regression datasets spanning five representative visual rating domains.

**Strengths:**

1. The submission proposes a coarse-to-fine method to improve existing MLLMs, and it shows performance gains.
2. The submission is clearly written, well-organized, and easy to follow.

**Weaknesses:**

1. STORM is essentially a merger of existing datasets without new annotations, which limits its contribution. The text annotations are not truly new, but just reformatted versions of existing numeric scores, so they do not add meaningful value. Overall, for a dataset-focused paper, the work mainly consists of combining prior datasets, which does not meet the expected level of novelty, effort, or complexity.
2. Despite fine-tuning, performance remains weak. Prior work (Q-Align, DeQA-Score) trained on discrete levels reports SRCC/PLCC above 0.9 on KonIQ, SPAQ, and ChallengeDB. In contrast, the proposed method only achieves SRCC or PLCC in the 0.6+ / 0.7+ range on IQA datasets (as shown in the Appendix), which is notably lower than many non-MLLM baselines such as MUSIQ, ManIQA, NIMA, HyperIQA, and DBCNN. This calls the effectiveness of the proposed method into question.
3. Converting numeric labels into fixed textual categories has been well-explored in Q-Align and DeQA, which is not novel. Simply increasing the number of quality levels does not fundamentally distinguish STORM from prior schemes, since those methods could also adjust the level number. A large level number may even harm generalization, so the key issue is choosing an appropriate trade-off rather than just adding more levels.
4. To convincingly claim a universal numeric regression scheme, the method should include a fully training-free version (or at least report test results without any task-specific training). Relying on SFT over a limited set of tasks still leaves generalization concerns, even if some tasks are evaluated in an OOD setting.

**Questions:**

My rating is mainly based on the limited dataset contribution, relatively poor performance, weakness of novelty, and weak support of the so-called "universal".

---

### Official Review · Reviewer_nBcA · 2025-11-01

**Soundness:** 3
**Presentation:** 3
**Contribution:** 1
**Rating:** 4
**Confidence:** 3

**Summary:**

The work presents a systematic benchmark framework for evaluating and enhancing the ordinal regression capabilities of MLLMs in visual scoring tasks. By introducing a coarse-to-fine reasoning mechanism, it aims to improve the subpar performance of MLLMs in these tasks.

**Strengths:**

The work constructs a dataset comprising 655,000 image-question pairs, encompassing the domains of image quality, aesthetics, age estimation, medical grading, and historical period estimation. Mainstream datasets in these areas often lack textual labels; thus, the authors employed GPT to generate labels, followed by manual adjustments to ensure that the textual definitions align more closely with human scoring habits.

**Weaknesses:**

1. Lacks of innovation: although the Coarse-to-Fine Chain-of-Thought approach is practical, this concept is common in IQA.
2. The experiments lack comparisons with SOTA models in the field, instead only comparing with other MLLMs, making it difficult to demonstrate its performance advantages in visual scoring tasks.
3. The work does not provide detailed information on the specific ratios and standards used for the manual adjustments in generating textual labels.
4. The work utilizes Qwen2.5-VL-3B as the backbone, but the results presented in the experiments do not show a significant improvement in performance.

**Questions:**

1. Why does the dataset of IQA exclusively select authentic distortion datasets, omitting synthetic distortion datasets?
2. In the Appendix, there is no detailed explanation for the reason why STORM-7B performs worse than STORM-3B, which is a particularly notable anomaly.

---

### Official Review · Reviewer_q1A7 · 2025-11-01

**Soundness:** 3
**Presentation:** 2
**Contribution:** 3
**Rating:** 4
**Confidence:** 4

**Summary:**

This paper introduces STORM, an ordinal regression dataset and benchmark designed to enhance the general visual scoring capabilities of MLLMs. STORM integrates 600k+ image-question-answer pairs from 14 ordinal regression datasets across 5 major visual scoring domains, and proposes a coarse-fine CoT prompting paradigm. This paradigm allows the model to first perform coarse-grained candidate filtering during inference before providing fine-grained interpretable scores, thereby mitigating digital illusion and improving cross-domain zero-shot performance. Experiments show that the 3B model fine-tuned based on STORM significantly outperforms baselines such as LLaVA-1.5-7B and Qwen2.5-VL-3B on both in-domain and out-of-domain tasks, validating the effectiveness of the framework and data.

**Strengths:**

1. This work gives the first comprehensive benchmark and dataset suite specifically designed to benchmark and boost the universal visual rating capability of MLLMs across 14 ordinal-regression datasets spanning five domains.

2. A coarse-to-fine Chain-of-Thought prompting pipeline that equips MLLMs with an interpretable, hierarchical ordinal-reasoning paradigm, substantially mitigating numerical hallucination and improving zero-shot generalization.

3. Experiment session is convincing, demonstrated that STORM establishes new state-of-the-art all-in-one and zero-shot performance on both in-domain and out-of-domain visual rating tasks, providing a solid foundation for future research on trustworthy MLLM-based scoring systems.

**Weaknesses:**

1. All textual labels rely on GPT-4 generation supplemented by manual fine-tuning, a process that is neither reproducible nor transparent. I understand the high cost of manual data, but this weakens the data contribution of this paper, making it resemble a "combination of existing datasets."

2. The zero-shot claim is mismatched with the evaluation settings. **The paper emphasizes "zero-shot" capability**, yet in most experiments, it uses the training set of the corresponding dataset for fine-tuning (even providing both lite and full scales). Only six test splits are retained for true zero-shot scenarios, resulting in insufficient robustness of the zero-shot performance claim.

3. Different domain tasks (such as IQA, IAA, FAE, MDG) are uniformly evaluated using accuracy and MAE, a mechanism that may not be reliable. Quality and aesthetics are qualitative factors (dividing into five levels is merely for model scoring convenience), while age is clearly quantitative. I am unsure whether dividing age into levels and using a multiple-choice-like evaluation method is reliable.

4. The compared models are all old, such as LLaVA-1.5 and InternVL, lacking new baselines.

5. The article seems to have been written in a hurry, with similar terms (such as MLLM and LMM) used interchangeably. It is recommended to check it.

**Questions:**

1. The conflict detection between the "intermediate thinking" and the final answer in coarse-fine CoT relies solely on a single prompt; a more comprehensive explanation is needed.

2. Unifying multiple tasks is indeed one of the contributions of this paper. However, is it reliable to divide all tasks into four/five tiers? The spacing between each tier is not equivalent. The authors should provide a detailed analysis of the alignment of each tier within the five task categories, such as 'bad quality' and '4-6 years old'. This seems illogical from a common-sense perspective.

3. More advanced MLLMs such as InternVL3 need to be considered.

---

### Note · Authors · 2025-11-14

I have read and agree with the venue's withdrawal policy on behalf of myself and my co-authors.